# Fetuin-A in Activated Liver Macrophages Is a Key Feature of Non-Alcoholic Steatohepatitis

**DOI:** 10.3390/metabo12070625

**Published:** 2022-07-07

**Authors:** Quentin Etienne, Valérie Lebrun, Mina Komuta, Benoît Navez, Jean-Paul Thissen, Isabelle A. Leclercq, Nicolas Lanthier

**Affiliations:** 1Laboratory of Gastroenterology and Hepatology, Institut de Recherche Expérimentale et Clinique, UCLouvain, 1200 Brussels, Belgium; quentin_etienne@hotmail.com (Q.E.); valerie.lebrun@yndigos.be (V.L.); isabelle.leclercq@uclouvain.be (I.A.L.); 2Service d’Anatomie Pathologique, Cliniques Universitaires Saint-Luc, UCLouvain, 1200 Brussels, Belgium; mina.komuta.pathology@gmail.com; 3Service de Chirurgie et Transplantation Abdominale, Cliniques Universitaires Saint-Luc, UCLouvain, 1200 Brussels, Belgium; benoit.navez@saintluc.uclouvain.be; 4Pole of Endocrinology, Diabetes and Nutrition, Institut de Recherche Expérimentale et Clinique, UCLouvain, 1200 Brussels, Belgium; jeanpaul.thissen@uclouvain.be; 5Service d’Endocrinologie, Diabétologie et Nutrition, Cliniques Universitaires Saint-Luc, UCLouvain, 1200 Brussels, Belgium; 6Service d’Hépato-Gastroentérologie, Cliniques Universitaires Saint-Luc, UCLouvain, 1200 Brussels, Belgium

**Keywords:** non-alcoholic steatohepatitis, metabolic dysfunction-associated fatty liver disease, fetuin-A, diabetes, insulin resistance, foz mice, humans, diabetes, liver, adipose tissue

## Abstract

Fetuin-A, a plasma multifunctional protein known to play a role in insulin resistance, is usually presented as a liver secreted protein. However, fetuin-A adipose tissue production has been also described. Here, we evaluated fetuin-A production by the liver and the adipose tissue during metabolic dysfunction-associated fatty liver disease (MAFLD)-non-alcoholic steatohepatitis (NASH) development. Fetuin-A was evaluated by enzyme-linked immunosorbent assay (ELISA), polymerase chain reaction (PCR), Western blot, and immunofluorescence in male foz^−/−^ mice fed a normal diet (ND) or a high fat diet (HFD) at various timepoints and in MAFLD-NASH patients. Foz^−/−^ mice fed a short-term HFD developed liver steatosis, insulin resistance, and increased circulating levels of fetuin-A compared to ND-fed mice. In mice and patients with NASH, fetuin-A was located not only in healthy or steatotic hepatocytes but also in some macrophages forming lipogranulomas. In both mice and humans, a significant amount of fetuin-A was present in the adipose tissue compared to the liver. However, messenger ribonucleic acid levels and cell culture experiments indicate that fetuin-A is produced by the liver but not by the adipose tissue. In conclusion, fetuin-A is produced by steatotic hepatocytes at early timepoints in MAFLD and correlates with insulin resistance both in mice and humans. In NASH, fetuin-A also co-localizes with activated liver macrophages and could be interpreted as a signal released by damaged hepatocytes.

## 1. Introduction

The visceral adipose tissue (AT), active in producing pro-inflammatory cytokines and releasing free fatty acids (FFA) into the blood stream, is considered as the main driver of type 2 diabetes and other metabolic complications [1]. However, growing evidence suggests that metabolic alterations within the liver induce a diabetogenic milieu earlier than the AT [2,3,4].

Metabolic dysfunction-associated fatty liver disease (MAFLD), defined by an increased amount of liver fat due to metabolic overload, is now the more frequent common cause of liver disease in developed countries and affects 30% of the population [5,6]. The prevalence of MAFLD is thus far higher than that of obesity, estimated at 12% of the population in Belgium [7]. MAFLD is associated with insulin resistance (IR) even in lean subjects [8,9]. IR involves the liver as well as distant organs and is independently associated with liver triglyceride content but not with waist circumference or BMI [8,9]. One could thus believe that the initial metabolic ill adaptation to Western dietary habits involves the liver. MAFLD is described as a wide spectrum of liver histological findings ranging from intracellular lipid accumulation into hepatocytes (steatosis) to harmful non-alcoholic steatohepatitis (NASH) encompassing steatosis, lobular inflammation, and hepatocyte ballooning that can further drive liver fibrosis [10]. In some cases, cirrhosis and hepatocellular carcinoma arise from NASH [10]. 

Change of the endoplasmic reticulum, involved in the creation and shaping of proteins, is one of the modifications occurring during MAFLD pathogenesis [11] with subsequent modification of the hepatic transcriptome, proteome, and secretome [12,13]. Among the proteins in which hepatic production is increased with MAFLD, attention has been given to fetuin-A [14], a hepatocyte secreted glycoprotein presented as a hepatokine [15]. Yet, animal and human data also suggest that adipocytes produce and secrete fetuin-A [16,17,18]. In healthy conditions, fetuin-A is involved in bone metabolism (mineralization, calcification inhibition), cardiovascular, and central nervous systems [19,20]. Fetuin-A plays also a key role in the inhibition of insulin signaling in the AT by phosphorylating the serine residues of insulin receptor substrate-1 (IRS-1), further conflicting with the physiologic tyrosine kinase activity of the insulin receptor on IRS-1 and therefore hampering the downstream pathways [21,22]. In addition, the release of fetuin-A from hepatocytes is a downstream effect of liver nuclear factor-kappa B (NFκB) pathway activation by FFA [23]. Fetuin-A also mediates the inflammatory role of FFA in the AT by presenting them to the toll-like receptor 4 (TLR4) and acting therefore as an endogenous ligand of TLR4 [14]. For these reasons, fetuin-A knockout (KO) mice demonstrate increased insulin sensitivity and decreased adiposity when fed a high fat diet (HFD) [24]. In humans and rodents, circulating levels of fetuin-A are elevated in the presence of liver steatosis [12,25,26] and higher fetuin-A level is a risk factor for developing type 2 diabetes [27]. However, up to now, no study compared the hepatic and AT fetuin-A production and the relative contribution of the production of each organ to the circulating levels of the protein in the context of NASH.

Macrophages from the AT [28,29] as well as those from the liver [3] are known to be active drivers of IR. In response to the sensing of signals from cellular damage/stress (damage associated molecular patterns), from the gut (such as pathogen-associated molecular patterns), or from overnutrition, macrophages switch to a pro-inflammatory phenotype and favor the recruitment and differentiation of blood monocytes into inflammatory cells observed in the liver and the AT of obese mice and individuals [30,31,32,33,34]. Therefore, depleting the chemokine macrophage chemoattractant protein-1 (MCP-1), also known as CCL2, significantly reduces IR [35]. Importantly, liver macrophages are implicated in human MAFLD pathophysiology at early timepoints (steatosis occurrence) but also at later timepoints of NASH development and aggravation [36]. Interestingly, fetuin-A participates in the chemoattraction of AT macrophages (ATM), to their subsequent polarization into the inflammatory phenotype and hence in the development of obesity-induced IR [18]. Nevertheless, to date, there are no data available exploring a potential interaction between fetuin-A and liver macrophages during MAFLD-NASH development.

Thus, here, we decided first to evaluate macrophage activation in MAFLD/NASH, then to characterize fetuin-A in the serum, the liver, and the AT during progression along the MAFLD spectrum and, finally, to investigate the possible interactions between fetuin-A and liver macrophages in this context. 

To do so, we used a mouse model of NASH, the foz^−/−^ mice under an HFD. Indeed, due to their deficiency for Alms1, a protein essential for primary cilium function, they are hyperphagic and prone to develop obesity and systemic IR together with progressive MAFLD from steatosis to fibrosing non-alcoholic steatohepatitis (NASH) [37,38,39]. We also performed cell culture experiments and took advantage of human samples (liver, AT, and serum) from obese MAFLD patients of our hepatology clinic to evaluate the production of fetuin-A by different tissues.

## 2. Results

### 2.1. Liver Steatosis Appears Early under HFD in Foz^−/−^ Mice and Progresses towards NASH

At the first timepoint (4 weeks), foz^−/−^ mice fed an HFD were characterized by liver steatosis. At this early timepoint, approximately 5% of hepatocytes showed lipid droplets mainly in the centrilobular region (Figure 1A). Steatosis was confirmed by a tenfold increase in liver lipid content (Figure 1B) compared to ND-fed animals. Notably, steatosis appears at this timepoint with no significant changes in body weight, liver weight, or glycemia (Figure 1C–E). However, mice had already a significant increase in insulin levels, consistent with IR (Figure 1F).

After 12 weeks of HFD feeding, steatosis had increased (Figure 1H), still located preferably in the centrilobular (zone 1) and mid-lobular (zone 2) hepatocytes, while periportal hepatocytes are mostly spared (Figure 1G). After 30 weeks of an HFD, liver steatosis invades the whole parenchyma, together with the appearance of inflammatory foci and hepatocyte ballooning, which are key features of steatohepatitis (Figure 1G). In parallel with MAFLD/NASH progression, mice develop obesity and hepatomegaly (Figure 1I,J). Hyperglycemia and hyperinsulinemia are at the highest level after 12 weeks of an HFD (Figure 1K,L). 

### 2.2. Macrophage Activation Occurs during MAFLD Progression

Because they are key players in IR and MAFLD progression, we therefore investigated macrophage activation. As evidenced by F4/80 immunochemistry on liver slides, macrophages which adopt an elongated and star-like shape are located in the sinusoids of the hepatic parenchyma in ND and HFD12W groups (Figure 2A). However, the image analysis software revealed no increase in the immunostained area in the liver of HFD4W and HFD12W mice compared to their ND counterpart (Figure 2B). A significant increase was highlighted after 30 weeks of an HFD (Figure 2B), together with macrophage crown-like structures (or lipogranulomas) (Figure 2A,C). This pattern is observed around fat-laden hepatocytes. Finally, macrophage activation was also characterized by the hepatic F4/80 gene expression analysis. There was a major 3-fold increase of F4/80 expression in HFD30W mice (Figure 2D). Interestingly, at 30 weeks of an HFD, monocyte-derived macrophage’s markers CD11b and CX3CR1 and inflammatory cytokine TNF-α were overexpressed compared to ND (Figure 2E–G).

### 2.3. Serum Fetuin-A Level Increases during Steatosis Progression 

We measured fetuin-A in serum samples by ELISA. Fetuin-A levels were increased significantly in HFD4W animals compared to ND animals and even more after 12 weeks of an HFD (Figure 3A). The level remains stable at 30 weeks (Figure 3A). Fetuin-A levels were positively correlated with blood glucose levels (Figure 3B). 

### 2.4. Characterization of Fetuin-A: Liver Versus Adipose Tissue Content and Production

Fetuin-A is described as a hepatokine. In disagreement with rising serum concentrations, hepatic protein and transcript levels were invariable with time (Figure 3C,D). By contrast, as detected by Western blot, fetuin-A protein content in AT increased after 12 weeks in foz^−/−^ mice (Figure 3E) and remained elevated at 30 weeks (not shown). Yet, no difference in transcript expression was evidenced upon HFD feeding (Figure 3F). Blood was used as a positive control. For a similar protein loading (serum and AT), the levels of fetuin-A were lower in the AT compared to the serum (Figure 3E). 

We then compared on the same gel liver and AT protein expression. A higher level of fetuin-A is consistently observed in the AT than in the liver in all groups (ND, HFD4W, HFD12W), even after normalization with an endogenous control (Figure 3G). Nevertheless, when normalized and compared in PCR, liver fetuin-A transcripts were 500- to 1000-fold higher in the liver than in the AT (Figure 3H). 

Collectively, our data point towards the expression and the production of fetuin-A by liver cells, its secretion in the blood stream, and variable trapping of the protein in the AT. To check this, we performed cell culture experiments: the supernatant of HepG2 cells does contain fetuin-A, at a higher level than within HepG2 cells themselves (cell lysate) (Figure 3I). However, there was no measurable fetuin-A in the supernatant of 3T3-L1 (not shown) or of 3T3-F442 adipocytes (Figure 3I), confirming that fetuin-A is produced by hepatocytes rather than by adipocytes. 

### 2.5. Histological Pattern of Fetuin-A during MAFLD/NASH Development

Further supporting its production by hepatocytes, in normal murine livers, fetuin-A is distributed as granules in the cytoplasm of hepatocytes, particularly in those around the centrilobular vein as shown by immunofluorescence studies (Figure 4A). After 4 or 12 weeks of an HFD, the lobular and cellular localization pattern is similar, and the fetuin-A protein is clearly visible in the cytoplasm around lipid droplets (Figure 4A). At 30 weeks of an HFD (Figure 4B), fetuin-A was again seen in (steatotic) hepatocytes but also in some macrophages (~10% of lipogranulomas), as evidenced by co-localization of the F4/80 (macrophage) and fetuin-A signals (Figure 4B). 

As the main function of macrophages is phagocytosis, F4/80 positive cells may have engulfed hepatocyte debris rich in fetuin-A. We performed a co-staining for fetuin-A and the macrophage lysosomal protein LAMP-1 confirming the co-localization of fetuin-A within endophagolysosomes in macrophages in the liver parenchyma of HFD30W animals (Figure 4C).

### 2.6. Validation of Results in Human MAFLD-NASH

With bariatric surgery, we had the unique chance to compare fetuin-A expression in the liver and the visceral AT. The analysis also revealed that the liver is the prominent site for fetuin-A transcript expression (Figure 5A). The protein is mainly found in the serum and tissue protein content is low (Figure 5B). Nevertheless, fetuin-A protein content is similar in liver and AT (Figure 5B).

As in the rodent model, glycaemia and serum fetuin-A concentrations were also positively correlated in humans (Figure 5C). Serum fetuin-A tended to be higher in patients with IR (HOMA-IR > 2.5) and in those treated for diabetes (Figure 5D) compared with patients without IR (HOMA-IR < 2.5), while there was also no difference according to the fibrosis stage (Figure 1E).

Similar to mouse NASH livers, positive CD68 lipogranulomas are also seen on liver biopsies of patients presenting NASH (Figure 5F). In human NASH, besides the intra-hepatocytes’ granular pattern of fetuin-A, dense fetuin-A aggregates within macrophages in inflammatory cell clusters and macrophages forming lipogranulomas (~50 to 100% of lipogranulomas) were also seen (Figure 5G).

## 3. Discussion

Accumulating evidence highlights that IR due to MAFLD is at the core of an interorgan hub in which immune cells, chemokines, and organokines are key effectors [30,34,40]. However, questions remain about the triggers driving the progression from simple steatosis to more severe steatohepatitis disease. In this study, we used a mouse model of MAFLD-NASH to investigate the production of fetuin-A in relation with macrophage activation. Our foz^−/−^ model developed similar metabolic (steatosis, IR, obesity) and histologic patterns as observed in human MAFLD/NASH patients. A rapid and significant increase of the circulating level of fetuin-A at the first 4-week timepoint compared to ND-fed mice is evidenced. This occurred in parallel with liver steatosis and increased circulating levels of insulin, but before any significant increase in body weight and blood glucose level. Elevated circulating glucose indeed appears later at 12 weeks, a timepoint when obesity and a more pronounced steatosis can be observed as well as the first signs of liver and AT inflammation due to macrophage activation. After 30 weeks of an HFD, liver disease is further aggravated [41]. Circulating fetuin-A did not further increase with disease progression to fibrosing NASH.

Our data indicate that fetuin-A is found at high concentrations in the serum. Indeed, when comparing the liver, the AT, and the serum by Western blot, we found that the protein was far more abundant in the circulating compartment compared with the two insulin sensitive tissues. Our results also support that fetuin-A protein accumulates in the AT during liver disease progression. This is intriguing since transcript expression is barely detectable and virtually invariant during model aggravation. However, this increase in fetuin-A in adipose tissue has already been demonstrated in adipose tissue of other animals than foz^−/−^ mice such as wild-type mice on a high fat diet [18], *db/db* mice [18], or in diabetic obese subjects [42]. This is compatible with fetuin-A being stored or trapped rather than produced in the AT. By contrast, in the liver, the level of transcript is high and AT transcripts are 500-fold lower. Together with a granular pattern of fetuin-A protein in hepatocytes, this supports that the protein is produced by hepatocytes and then exported to the blood stream. In support, cell culture experiments clearly identified hepatocytes (HepG2) as fetuin-A producing and secreting cells, while cultured differentiated adipocytes 3T3-L1 and 3T3-F442A failed to express and produce. Similarly, researchers failed to detect fetuin-A protein in adipocyte cultures and supernatant (except when brought through the system by the addition of FBS) and transcript was undetectable [42]. 

Human data on MAFLD/NASH patients support a similar operation than in the mouse model. Indeed, we showed a correlation between fetuin-A and blood glucose levels. Unfortunately, a control group was missing; as for ethical reasons, liver biopsies are not performed in this population of control subjects. The MAFLD subgroup without IR was also very small, as steatosis that occurs in a dysmetabolic context is very often associated with IR. When compared to fetuin A levels in control subjects reported in other studies (<500 µg/mL) [15,27], levels observed in MAFLD patients are significantly higher (>500 µg/mL). Thus, collectively, mice and human data are consistent with increased circulating fetuin-A at the early phase of IR with no further increase with disease progression. These data are in line with a recent large meta-analysis of 1755 MAFLD patients and 2010 control patients [43]. The researchers showed an increase in fetuin-A levels in MAFLD patients compared to controls, but no significant change according to disease severity (NASH or presence of fibrosis) [43].

Another real breakthrough of this study is the immunofluorescence data illustrating the localization of the protein in the hepatic parenchyma during liver disease, which is, to our knowledge, the first description. While fetuin-A was located in perivenular hepatocytes in healthy control mice, its expression was then evidenced mainly in steatotic hepatocytes of mice fed the HFD for 4 and 12 weeks. We found then a specific signature of NASH (in foz^−/−^ mice as in humans) where fetuin-A was not only located in steatotic hepatocytes but also in activated macrophages (crown-like structures and clusters). Indeed, our data demonstrated progressive macrophage activation starting with slight increase of macrophage marker F4/80 expression at 12 weeks of the HFD with only some mice exhibiting macrophage “crown-like structures” in liver parenchyma, whereas at 30 weeks of the HFD, a threefold increase in F4/80 expression with a higher F4/80 stained area and numerous crown-like structures and inflammatory clusters. In human patients suffering from NASH, similar crown-like structures also arise [44]. Those data suggest an interaction between fetuin-A and macrophages in advanced disease stage. Kupffer cells are key to clear cell debris upon hepatocyte death. Our results show co-localization of intense fetuin-A staining with the lysosomal protein LAMP-1 in lipogranuloma compatible with endocytoses of fetuin-A. Hypothetically, after engulfment of fat-laden dying hepatocytes, trapped fetuin-A can link with lipid residues triggering a macrophage pro-inflammatory response that further sustains lobular inflammation and progression of NASH. Thus, our data, as well as data from other researchers who have demonstrated the presence of fetuin-A protein into macrophages from obese patients in the absence of fetuin-A mRNA, is compatible with the phagocytosis of fetuin-A by macrophages [42]. These new outcomes uncover new perspectives to explore fetuin-A within activated macrophages as a histological signature for severe NASH as well as unveil the mechanistic consequence of fetuin-A within liver macrophages in the progression of NASH. 

## 4. Materials and Methods 

### 4.1. Animals and Diet

Foz^−/−^ (Alms1 mutant) mice with NOD B10 background were bred and maintained in a 12 h light/dark cycle with permanent access to food and water. After weaning, mice were fed a normal diet (ND) containing 2.83 kcal/g (16% lipids, 54% carbohydrates, 30% proteins, 0.001% cholesterol; A03 from SAFE-diets) or a high fat diet (HFD) containing 5.24 kcal/g (60% lipids, 20% carbohydrates, 20% proteins, 0.03% cholesterol; D12492 from Research Diets) for 4, 12, and 30 weeks. The experiments were carried out in two stages: a first set of animals was given the HFD (or ND) for 4 weeks (HFD4W) and a second set of animals received the HFD (or ND) for 12 (HFD12W) or 30 weeks (HFD30W). The experiments were carried out on a minimum of 4 animals per group. The animals were handled according to the guidelines for humane care for laboratory animals established by the UCLouvain in accordance with European regulation, and the study protocol was approved by the university ethics committee (2008/UCL/MD/016 and 2012/UCL/MD/026). At the time of euthanasia, mice were anesthetized (ketamine/xylazine 1:1) and intracardiac blood collection induced the death of animals, and then, liver and AT were collected. One part of the tissues was fixed in formaldehyde 4% for histological analysis and the other part was placed into liquid nitrogen and stored at −80 °C for subsequent analysis. 

### 4.2. Human Data

Two human cohorts were studied. The translational project was approved by the hospital-faculty ethics committee (2014/15OCT/514) and patients agreed to participate by signing an informed consent after receiving information.

The first cohort included 49 patients in whom a liver biopsy was performed for MAFLD diagnosis evoked by several supportive parameters: obesity or metabolic syndrome, increased liver transaminases, liver echography, and controlled attenuation parameter (CAP) suggesting steatotic liver disease. Liver biopsy confirmed MAFLD or NASH in all patients. Other common causes of chronic liver diseases were excluded. Fasted blood was sampled to assess glycemia, insulinemia, and fetuin-A.

The second cohort included 5 obese patients who underwent bariatric surgery and had a suspicion of MAFLD based on blood results exhibiting increased transaminases, after exclusion of other causes of chronic liver disease. Fasted blood collection was performed before surgery for the measurement of glycemia, insulinemia, and fetuin-A. At the time of surgery, laparoscopic liver biopsy (Bard Monopty) and a sample of omentum (visceral AT) were collected by the surgeon. 

A part of the liver biopsy (and also AT sample in the case of bariatric surgery) was placed in formaldehyde for histological analysis, and two other parts in RNA stabilization solution (RNALater) and liquid nitrogen for transcriptomic and proteomic analysis, respectively. 

### 4.3. Cell Culture Experiments

The human hepatocellular cell line (HepG2) was cultured at 37 °C with 5% CO_2_ in high-glucose DMEM containing 10% (*v/v*) fetal bovine serum (FBS), 100 units/mL penicillin, and 100 μg/mL streptomycin (P/S) in a 75 cm^2^ flask (Nunc™ Cell Culture Treated EasYFlasks™, T75, filter—Thermo Fisher, Waltham, MA, USA) with fresh medium every 48 h until confluency. Cells were then cultured in 6-well plates in the same medium.

The murine fibroblasts 3T3 are adipocyte precursors. Two different cell lines (3T3-L1 and 3T3-F442A) were cultured with differentiating medium containing IBMX (3-isobutyl-1-methylxanthine), dexamethasone, insulin, and rosiglitazone as described elsewhere to generate adipocytes [45]. Briefly, subcultured cells were cultured and maintained at confluency for 48 h in Basal Medium I (DMEM + glutamine + pyruvate + PS + 10% New-born Calf Serum (NCS)). Differentiation was then induced by incubating the cells with the Differentiation Medium I (DMEM + glutamine + pyruvate + FBS + P/S + 0.5 mM IBMX + 0.25 μM dexamethasone + 1 μg/mL insulin +2 μM de rosiglitazone). After 48 h, the cells were incubated in Differentiation Medium II (DMEM + pyruvate + glutamine + PS + 10% FBS + insulin 1 μg/mL) for 2 days. Once differentiated, the cells were maintained in Basal Medium II (DMEM + glutamine + pyruvate + PS + 10% FBS) for further analyses. To assess the expression of fetuin-A by cells and its concentration in the supernatant, cells are incubated in DMEM without FBS for 48H before harvesting the cells and supernatants for further analysis.

### 4.4. Evaluation of Liver Lipid Content

Total liver lipids were extracted with the methanol and chloroform method and quantified by the vanillin-phosphoric acid reaction as described elsewhere [3].

### 4.5. Histology, Immunohistochemistry, and Morphometrical Analysis

Immunohistochemical detection of liver macrophages was performed on paraffin-embedded sections (4 µm) treated with the antigen retrieval reagent proteinase K and using a primary rat anti-mouse F4/80 monoclonal Ab (1/200 MCA497G—Biorad Laboratories, Hercules, CA, USA), a rabbit anti-rat immunoglobulin (1/100, Vector Laboratories, Burlingame, CA, USA), and then a goat anti-rabbit streptavidin horseradish peroxidase-conjugated Ab (EnVision, Dako, Agilent, Santa Clara, CA, USA). In human tissue, liver macrophages were evidenced on paraffin-embedded sections treated with proteinase K and using a primary mouse anti-human CD68 antibody (Santa Cruz Biotechnoloy, Dallas, TX, USA) and then an anti-mouse streptavidin horseradish peroxidase-conjugated Ab (EnVision, Dako, Agilent, Santa Clara, CA, USA). The peroxidase activity was revealed with diaminobenzidine (DAB) and slides were counterstained with hematoxylin.

Double immunofluorescence was used to detect liver macrophages and fetuin-A on paraffin-embedded sections (4 µm) treated with proteinase K. A solution containing the 2 primary antibodies was applied on slides: rat anti-mouse F4/80 monoclonal Ab (1/200, MCA497G—Biorad Laboratories, Hercules, CA, USA) and goat anti-fetuin-A (1/100, M-17: sc-9668—Santa Cruz biotechnology, Dallas, TX, USA) for mouse samples or mouse anti-human CD68 antibody (1/200, Santa Cruz Biotechnology, Dallas, TX, USA) and goat anti-fetuin-A (1/100, M-17: sc-9668, Santa Cruz Biotechnology, Dallas, TX, USA) for human tissues. Secondary antibodies (Alexa Fluor donkey anti-goat 488 (for fetuin-A), donkey anti-rat 594 (for F4/80), or donkey anti-mouse 594 (for CD68) as appropriate) and the nucleus dye DiAmidino Phenylindole 1/5000 (DAPI) were then applied on the section.

For the detection of lysosomal protein LAMP-1, mice liver sections were treated in a citrate bath for antigen retrieval followed by permeabilization in PBS/Triton X-100 0,3%. Primary antibodies (1/100) rat anti-LAMP-1 (1D4B—DHSB) and goat anti-fetuin-A (M-17: sc-9668—Santa Cruz biotechnology) were used and then antigens revealed with secondary antibodies Alexa Fluor donkey anti-rat 564 (for LAMP-1) and Alexa Fluor donkey anti-goat 488 (for fetuin-A).

To quantify macrophage content, slides immunostained for F4/80 were scanned (Leica) and computationally analyzed (Visiopharm^®^) to quantify the stained area of DAB, the number of macrophage structures called lipogranulomas (or crown-like structures), and macrophage clusters. Algorithms were designed to individually detect and count liver macrophages (positive area) or macrophages forming lipogranulomas and clusters (positive area and number). 

### 4.6. Fetuin-A Serum Concentration

Serum fetuin-A quantification was performed using enzyme-linked immunosorbent assay (ELISA) kits from Quantikine—R&D Systems following the protocols detailed by the manufacturer (MFTA00 for mice, DFTA00 for humans). 

### 4.7. Protein Studies: Western Blotting

Liver and AT homogenates were prepared as described elsewhere [46] and protein concentration measured after colorimetric detection (Pierce™ BCA Protein Assay Kit) on a spectrophotometer at a 562 nm wavelength. Proteins were assayed by Western blotting. The immunoreactivity was detected with a horseradish peroxidase-conjugated secondary Ab (1/10,000) and chemiluminescence (Western Lightning Chemiluminescence Reagent Plus; Perkin Elmer, Waltham, MA, USA). One membrane was sequentially probed with multiple antibodies if appropriate (different molecular weight) and actin to control for protein loading. The quantification of immunoreactive proteins was obtained by densitometry using the Quantity One device and software (Bio-Rad Laboratories, Hercules, CA, USA). The levels of immunoreactivity relative to the invariant control are reported as arbitrary densitometry units.

### 4.8. RNA Extraction and PCR Analysis

Total RNA was extracted from frozen liver and epididymal fat samples using TRIpure Isolation Reagent (Roche Diagnostics Belgium, Vilvoorde, Belgium). cDNA synthesis and real-time PCR analysis were carried out as previously described. Primer pairs for transcripts of interest were designed using the Primer Express software (Applied Biosystems, Lennik, Belgium). RPL19 mRNA was chosen as an invariant standard. All experimental tissues and standard curve samples were run in duplicate in a 96-well reaction plate (MicroAmp Optical, Applied Biosystems). Mouse primer sequence for fetuin-A was TGGCCTGCAAGTTATTCCAAA (forward) and GCTGTGGGTACGGGACCTACT (reverse). Mouse primer sequence for CX3CR1 was ATCAGCATCGACCGGTACCT (forward) and CTGCACTGTCCGGTTGTTCAT (reverse). Human primer sequence for fetuin-A was GCACGCCGCGAAAGC (forward) and TTCCTCCAGCTGAAAATTGGA (reverse). The sequences of the other primers (F4/80, TNF-α, CD11b, CD68, and RPL-19) have already been described [3,46]. Results are expressed as fold expression relative to expression in the control group following the ΔΔCt method.

### 4.9. Statistical Analysis

Statistical analyses were carried out using GraphPad Prism^®^ for Windows (v.6.01, GraphPad Software, La Jolla, CA, USA). Data are presented as mean ± standard deviation. For comparison between two groups, parametric two-tailed unpaired Student’s T test was applied in cases of normal distribution (Shapiro–Wilk) equality of variances (F Test). When distribution was normal and variances were significantly different, an unpaired T test with Welch correction was applied. Alternatively, a non-parametric Mann–Whitney test was used in cases of non-normal distribution.

Comparison of more than two groups was performed with a parametric ANOVA test followed by multiple comparisons with the mean of every other column. In cases of equality of variance, Tukey’s multiple comparisons test was applied, or otherwise, Dunnett’s multiple comparisons test was used. Alternatively, in cases of non-normal distribution, non-parametric Kruskal–Wallis followed by Dunn’s multiple comparison tests were applied. For correlation analysis, Pearson’s correlation coefficient between pairs was applied in cases of normal distribution. Alternatively, Spearman’s correlation was used. We considered *p* ≤ 0.05 to be statistically significant. 

## Figures and Tables

**Figure 1 metabolites-12-00625-f001:**
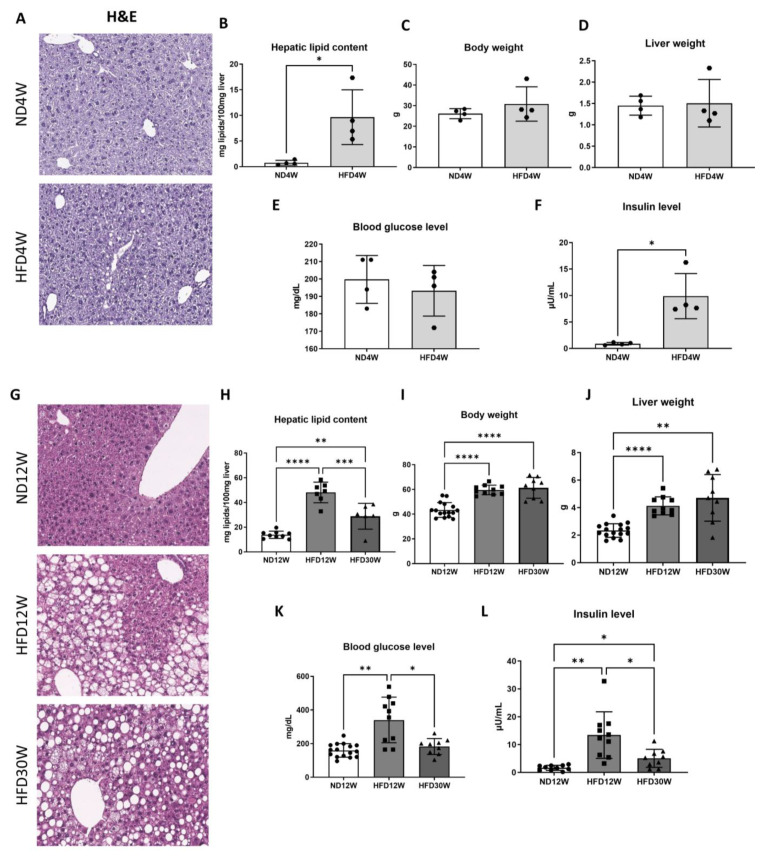
Foz^−/−^ mice develop steatohepatitis, obesity, and insulin resistance. Liver haematoxylin and eosin staining (H&E) (**A**), hepatic lipid content (**B**), body weight (**C**), liver weight (**D**), blood glucose level (**E**), and insulin level (**F**) in mice fed a normal diet (ND4W) or a high fat diet for 4 weeks (HFD4W). Liver haematoxylin and eosin staining (H&E) (**G**), hepatic lipid content (**H**), body weight (**I**), liver weight (**J**), blood glucose level (**K**), and insulin level (**L**) in mice fed a normal diet (ND12W) or a high fat diet for 12 (HFD12W) or 30 weeks (HFD30W). *n* = 4–12/group. * *p* < 0.05, ** *p* < 0.01, *** *p* < 0.001, **** *p* < 0.0001.

**Figure 2 metabolites-12-00625-f002:**
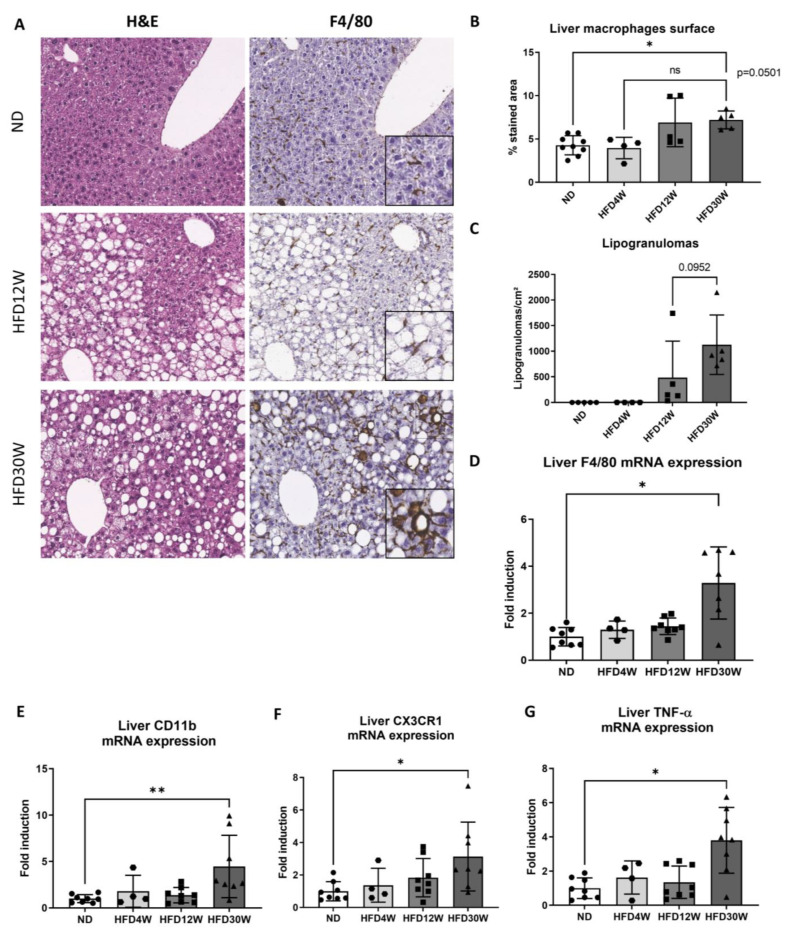
Foz^−/−^ mice show activation of liver macrophages with the appearance of crown-like structures. Liver haematoxylin and eosin staining (H&E) and F4/80 macrophage marker in mice fed a normal diet (ND) or a high fat diet for 12 (HFD12W) or 30 weeks (HFD30W). The insets in the image show enlargements centred on the macrophages (**A**). Quantification of liver positive F4/80 macrophage surface on liver slides (**B**) and of crown-like structures or lipogranulomas (**C**). Liver mRNA expression of F4/80 (**D**), CD11b (**E**), CX3C chemokine receptor 1 (CX3CR1) (**F**), and tumor necrosis factor alpha (TNF-α) (**G**). *n* = 4–8/group. * *p* < 0.05, ** *p* < 0.01.

**Figure 3 metabolites-12-00625-f003:**
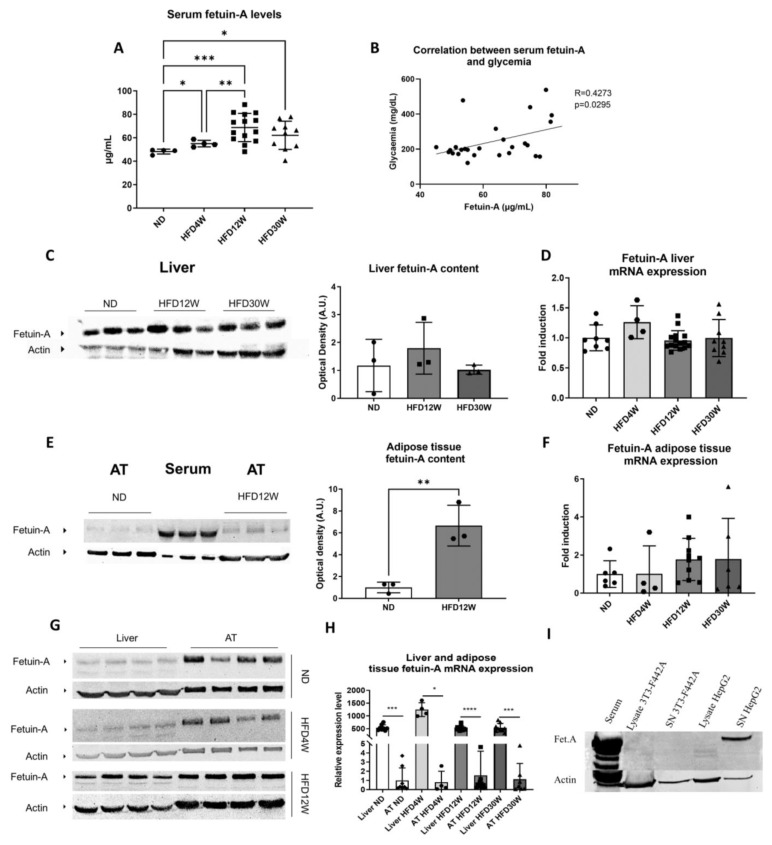
Elevation of fetuin-A levels in relation to insulin resistance and characterization of levels in serum, liver, and adipose tissue in mice. Serum fetuin-A levels in normal diet (ND) and high fat diet mice for 4 (HFD4W), 12 (HFD12W), or 30 weeks (HFD30W). *n* = 4–12/group (**A**). Correlation between serum fetuin-A levels and blood glucose levels (**B**). Western blots and corresponding quantifications based on averages of actin (endogenous control) corrected fetuin-A labelling intensity ratios for liver (**C**–**G**), adipose tissue (AT), and serum (**E**,**G**). Expression of fetuin-A (messenger RNA) in liver (**D**,**H**) or adipose tissue (**F**,**H**) of foz^−/−^ mice on normal (ND) or high fat diet for 4, 12, or 30 weeks (HFD4W, HFD12W, HFD30W). *n* = 4–12/group. * *p* < 0.05, ** *p* < 0.01, *** *p* < 0.001, **** *p* < 0.0001. Fetuin-A content in lysates and supernatants (SN) of 3T3-F442A (adipocytes) or HepG2 (hepatocytes) cells (**I**).

**Figure 4 metabolites-12-00625-f004:**
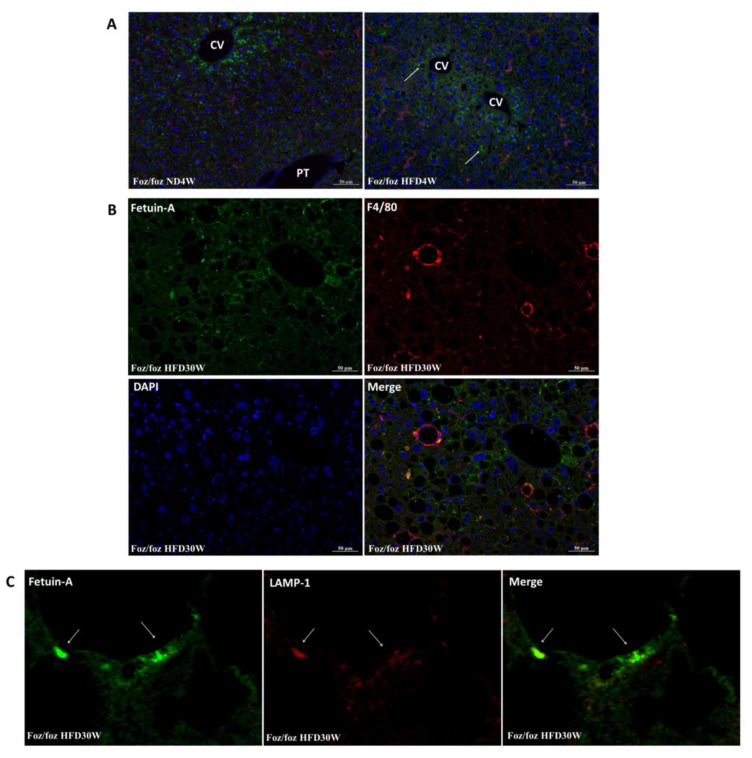
Fetuin-A is present in hepatocytes but also in activated macrophages in steatohepatitis. Immunofluorescence detection of fetuin-A (green) and hepatic macrophages (red) in a healthy liver of a foz^−/−^ mouse on a normal diet for 4 weeks (ND4W) or a high fat diet for 4 weeks (HFD4W). CV: centrilobular vein, PT: portal tract, arrow indicates concentration of fetuin-A in the cytoplasm of steatotic hepatocytes (**A**). Immunofluorescence detection of fetuin-A (green) and hepatic macrophages (red) representative of the group of mice fed a high fat diet for 30 weeks (HFD30W) with fetuin-A positive lipogranulomas (**B**). Immunofluorescence of fetuin-A (green) and lysosomal protein LAMP-1 (red) in the liver of a foz^−/−^ mouse representative of the high fat diet group (HFD30W). Image centered on a cluster of fetuin-A collocated with LAMP-1 (**C**).

**Figure 5 metabolites-12-00625-f005:**
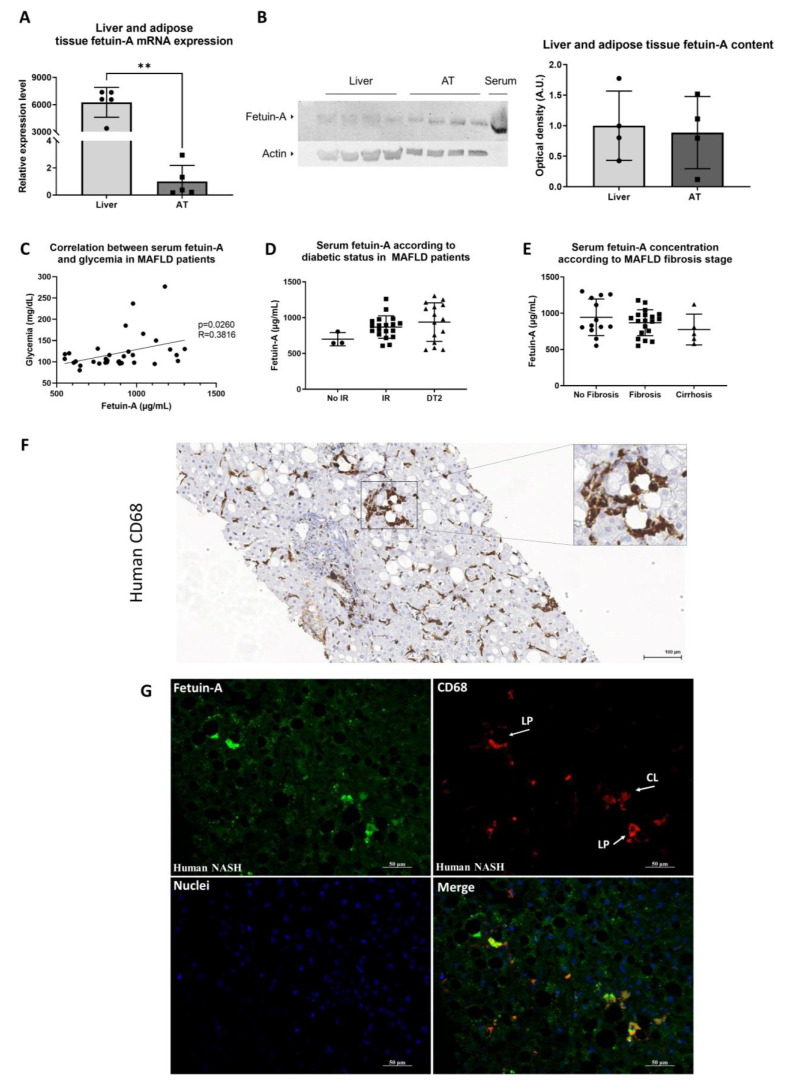
Characterization of fetuin-A in human metabolic dysfunction-associated fatty liver disease. Comparison of liver and adipose tissue (AT) expression of fetuin-A (mRNA) in humans with metabolic dysfunction-associated fatty liver disease (MAFLD). RPL19 served as endogenous control (*n* = 5/group), ** *p* < 0.01 (**A**). Western blot comparison of liver, adipose, and serum fetuin-A content. The corresponding quantification is calculated on the basis of the averages of the ratios of the labelling intensities of fetuin-A corrected by those of actin (endogenous control) (*n* = 4/group) (**B**). Correlations between serum fetuin-A levels and glycaemia in MAFLD patients (**C**). Serum fetuin-A levels in MAFLD patients: HOMA < 2.5 (*n* = 3), HOMA > 2.5 (*n* = 18), or patients treated for type 2 diabetes (DT2) (*n* = 16) (**D**). Serum fetuin-A levels in MAFLD patients with no fibrosis (*n* = 13), fibrosis (*n* = 20), or cirrhosis (*n* = 5) (**E**). Immunohistochemistry for the macrophage marker CD68 on histological section of liver biopsy from a patient with non-alcoholic steatohepatitis (NASH). Activated macrophages in the form of clusters and lipogranulomas are visible (inset) (**F**). Immunofluorescence of fetuin-A (green) and CD68 macrophages (red) in a human NASH liver section. Presence of lipogranulomas (LP) and one cluster of macrophages (CL) also positively labelled for fetuin-A (**G**).

## Data Availability

The data that support the findings of this study are available from the corresponding author upon reasonable request.

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
