# Peer review of "Fetuin-A in Activated Liver Macrophages Is a Key Feature of Non-Alcoholic Steatohepatitis"

_metabolites, 2022, doi:10.3390/metabo12070625_

Round 1

Reviewer 1 Report

In this report Etienne et all investigate the expression of fetuin A in Foz-/- mice fed a short and long term HFD. They report that Fetuin A protein is present in both in adipose and hepatic tissue, but the mRNA levels are multiple fold higher in liver. They suggest that Fetuin A should be secreted by liver and taken up by AT.

Major issues.

1)      In order to conclude that AT expression is playing a major in AT fetuin accumulation, the authors should investigate AT fetuin A accumulation in hepatic or adipose tissue specific Fetuin A KO mouse. Without genetic model this claim is speculation and not well supported by the presented data.

2)      In their abstract they state that fetuin A is produced by steatotic hepatocytes, however their data shows that fetuin A is expressed in normal and healthy hepatocytes as well. Furthermore, they claim that fetuin A co-localizes with F4/80. However, there is no quantification presented. The authors should show some type of quantification and statistics, as a significant portion of their discovery relies on this co-localization.

3)      The western blot quantification has to be improved as the quantification shown in bar graphs and the ( figure 3 C) WB picture are confusing. For example, the actin band in Normal chow fed mice looks to be at least couple of times less than the actin bands in 30 week HFD fed animals with little change in Fetuin A band, yet the quantification in the graph shows similar levels of Fetuin A/actin ratio.

4)      There is no information for the n of samples, furthermore, the data is shown as solid bar graphs, all data should be shown as in 3a, bars with individual points visible.

5)      No primer sequences are provided.

Author Response

1) Response to the first comment:

Thank you for your comment. We do not conclude that the AT expression plays a role in the accumulation of fetuin-A within it. We believe that fetuin-A produced by the liver is then secreted into the blood and stored in the adipose tissue.

The two following sentences are taken from the text to explain this:

“Collectively, our data point towards the expression and the production of fetuin A by liver cells, its secretion in the blood stream and variable trapping of the protein in the AT.”

“This is intriguing since transcript expression is barely detectable and virtually invariant during model aggravation. This is compatible with fetuin-A being stored or trapped rather than produced in the AT. By contrast, in the liver, the level of transcript is high and AT transcripts are 500-fold lower.”

These original data come from the results of Western blot (high amount of fetuin-A protein) and PCR (low amount of fetuin-A mRNA). See figure 3G and 3H.

Interestingly, other researchers are currently evaluating another hepatokine called fetuin-B and have presented their results at the European Fatty Liver Congress recently (EFLC 2022), reaching the same conclusions (Pasmans K and Meex R « The role of the hepatokine fetuin B in glucose homeostasis and adipose tissue function »).

KO mice for fetuin-A have already been studied. A sentence has therefore been added to the introduction:

"For these reasons, fetuin-A knockout (KO) mice demonstrate increased insulin sensitivity and decreased adiposity when fed a high fat diet (HFD) (ref: Mathews, Diabetes)."

It is certain that being able to create KO mice for fetuin-A specifically in the liver (and in hepatocytes) or in adipose tissue would be extremely interesting. However, this is very complicated. This takes much longer than the time given for the response to reviewers. We plan to work on this but the results would not be available for a few years. In the meantime, we think that our data are sufficiently robust and interesting and give indirect arguments for hepatic production of fetuin-A and storage in adipose tissue. Data from cell culture experiments are also strong to support this hypothesis (Figure 3I).

2) Thank you for your very justified comment. It is true that fetuin-A is produced by steatotic hepatocytes but also by healthy hepatocytes in mice on normal diet. We have therefore modified the sentence in our abstract to be absolutely correct.

"In mice and patients with NASH, fetuin-A was located not only in healthy or steatotic hepatocytes but also in some macrophages forming lipogranuloma."

Quantification of double-positive structures involves scanning of immunofluorescence images, which is complex. We therefore decided to perform a scan and accurate quantification of lipogranulomas. This quantification is presented in Figure 2C and shows indisputably the absence of lipogranulomas on a normal diet and their appearance after 12 and 30 weeks. Only when we observe the presence of lipogranulomas in Foz-/- mice do we observe the presence of fetuin-A within the macrophages (Figure 4B). The quantification with the statistical tests to be used is therefore presented in Figure 2C.

3) Western blot quantification was performed using a software and obtained by densitometry using the Quantity One device and software (Bio-Rad), as in our previous work (Lanthier N et al. Am J Physiol. 2010 and Lanthier N et al. FASEB J. 2011). We have carefully checked all our values. It is important to note that each value depends on the intensity of the spot but also on the background (which is subtracted). Figure 3C therefore corresponds to the values measured with the program described.

4) Thank you for your comment. We now clearly indicate in the materials and methods the minimum number of animals per group.

"The experiments were carried out on a minimum of 4 animals per group."

We have also modified all our graphs and placed the results as individual points in all our figures. In this way, the number of animals per group is clearly identifiable (see new Figures 1,2,3,5).

5) Sorry for the error. We have added the information in the materials and methods.

Reviewer 2 Report

This study evaluated fetuin-A production by the liver and the adipose tissue during MAFLD-NASH development, providing a potential prognosis marker.

1.     Figure 2A. Please add the zoom-in picture to show the morphology of the macrophage in ND and HFD12W groups.

2.     Figure 2G. Please correct the subtitle. TNF-a was not displayed appropriately.

3.     Figure 4C. Please add the F4/80 panel to show the macrophage.

Author Response

1) Response to the first comment:

This has been added and modified.

2) This has been corrected.

3) Unfortunately, this was not possible. As explained in the materials and methods, the antibodies against F4/80 and LAMP-1 that were used are both rat antibodies. It was therefore not possible to perform immunofluorescence labelling for F4/80 and LAMP-1 on the same section (same fluorescent anti-rat secondary antibody). However, we have chosen to show an exemplary case with a typical image of fetuin-A as an aggregate as typically seen in F4/80 positive macrophages (see Figure 4B).

Reviewer 3 Report

The article submitted by Lanthier and coworkers is really suitable for publication in this journal. The paper describe the behaviour and the role of fetuine A in the serum considering its involvement into insuline resistance. The metabolic dysfunction-associated fatty liver disease, the related macrophage activation in this disease, and the presence of fetuin-A in the serum were investigated establishing a putative interaction between fetuin-A and liver macrophages. 

The paper is well written, the speech is fluid and also the English language is correct.

Minor comments. Each sub-paragraphs need to be numerated and not labeled by a black dot.

Author Response

Response to the comment:

Thank you for your positive comments and feedback on our article.

We have changed the paragraphs and numbered them as requested.

Round 2

Reviewer 1 Report

The authors have improved the manuscript significantly. However, there are couple of issues that should be addressed.

1) the authors justification that on the western blot quantification is not satisfactory( Figure 3C).  I thank the author for explaining how the western blot quantification works,  however, for last 3 AT samples it seems that there might be depletion of substrate due high amount of HPR, which could distort the quantification. Given the high-reliance of the manuscript on this data, this WB should be repeated, preferably with higher number of samples. 

2) While the double positive structures  might be complex, it will be more more convincing to show some type of quantification for Fetuin A positive lipogranulomas. Otherwise, it might given an impression that the authors are overly relying on a single double- positive image. 

3). The authors should clearly state the n of samples in each experiment. 

Minor, the authors refer to " our country", naming the country will be better option. 

Author Response

Thank you for your comments. We have responded point by point (see attached file) and modified the text accordingly.
